# An investigation into gender distributions in scholarly publications among dental faculty members in Iran

Ahmad Sofi-Mahmudi[1,2,3]*, Erfan Shamsoddin[3], Lisa M. DeTora[4], Barbara E. Bierer[5], Perihan Elif Ekmekci[6], Morenike Oluwatoyin Folayan[7], Ching Shan Lii[8], Marcos Roberto Tovani-Palone[9], Francis P. Crawley[10]

1 Department of Health Research Methods, Evidence and Impact, McMaster University, Hamilton, Ontario, Canada, 2 Department of Anesthesia, National Pain Centre, McMaster University, Hamilton, ON, Canada, 3 National Institute for Medical Research Development (NIMAD), Cochrane Iran Associate Centre, Tehran, Iran, 4 Department of Writing Studies and Rhetoric, Hofstra University, Hempstead, NY, United States of America, 5 Department of Medicine, Division of Global Health Equity, Brigham and Women's Hospital, Harvard Medical School, Boston, Massachusetts, United States of America, 6 Department of History of Medicine and Ethics, TOBB ETU University Medical School, Ankara, Turkey, 7 Department of Child Dental Health, Obafemi Awolowo University, Ile-Ife, Nigeria, 8 Department of Pharmacy, Kuala Lumpur General Hospital, Kuala Lumpur, Malaysia, 9 Ribeirão Preto Medical School, University of São Paulo, Ribeirão Preto, Brazil, 10 Good Clinical Practice Alliance-Europe (GCPA) & Strategic Initiative for Developing Capacity in Ethical Review (SIDCER), Leuven, Belgium

* sofima@mcmaster.ca, a.sofimahmudi@gmail.com

**Data Availability Statement:** The data underlying the results presented in the study are available

## Abstract

### Background

Research on gender inequality is crucial as it unveils the pervasive disparities that persist across various domains, shedding light on societal imbalances and providing a foundation for informed policy-making.

### Aim

To investigate gender differences in scientometric indices among faculty members in dental schools across Iran. This included overall data and speciality-specific data.

### Methods

The publication profiles of academic staff in all dental schools were examined using the Iranian Scientometric Information Database (ISID, http://isid.research.ac.ir). Variables analyzed were working field, academic degree, the total number of papers, papers per year, total number of citations, percentage of self-citation, h-index, g-index, citations per paper, gender, university type, number of years publishing, proportion of international papers, first-author papers, and corresponding-author papers. Mann-Whitney and Kruskal-Wallis non-parametric tests were used to analyze the relationship between background characteristics and scientometric indicators. The extracted data were analyzed using R v4.0.1.

from our figshare repository at https://doi.org/10.6084/m9.figshare.23669190.

**Funding:** The author(s) received no specific funding for this work.

**Competing interests:** The authors have declared that no competing interests exist.

## Results

The database included 1850 faculty members, of which about 60% (1104 of 1850) were women. Men (n = 746) had a higher number of papers (6583 vs. 6255) and citations (60410 vs. 39559) compared with women; 234 of the 376 faculty members with no papers were women. Almost half of the women (N = 517 of 1104) were in Type 2 universities, and nearly half of the men (N = 361 of the 746) were faculty members at Type 1 universities (Type 1 universities ranking higher than Type 2 and 3 universities). The medians of scientometric indices were higher in men, except for self-citation percentage (0 (IQR = 2) vs. 0 (IQR = 3), P = 0.083), international papers percentage (0 (IQR = 7.5) vs. 0 (IQR = 16.7), P<0.001). The proportion of corresponding-author papers was more than 62% higher in women (25 (IQR = 50) vs. 15.4 (IQR = 40), P<0.001). Men had a two-fold higher median h-index (2 (IQR = 4) vs. 1 (IQR = 3), P<0.001). Restorative dentistry and pediatric dentistry had the highest men-to-women ratios (1.5 for both). Dental materials and oral and maxillofacial surgery showed the lowest men-to-women ratios (0.42 and 0.5, respectively).

## Conclusions

Women made up the majority of dental faculty members in Iran. Nevertheless, men showed better scientometric results in several significant indices. Having identified scientometric information reflecting differences across faculty members, further research is now needed to better understand the drivers of these differences.

## Background

In recent decades, gender equality (as a part of social equality) has been increasingly sought globally in the medical sciences. The United Nations links greater gender equality with economic development and prosperity as well as being responsible for a significant share of new economic growth over the past fifty years attributable to greater educational access among girls [1,2]. Gender equality is also seen as a contributing factor to sustainable development [3]. Science and technology fields have been less successful in achieving gains toward gender equality than found in other professional areas [4]. Skewed application processes, poor mentorship opportunities, and social pressures (such as disproportionate household responsibilities) are suggested to contribute to this problem [4–8]. Despite the higher rate of female undergraduate and graduate students in many countries [9], there are fewer female PhD candidates, postdocs, and professors [6,10]. Even for women who gain entry to the sciences, barriers may exist to achieving scientific publications, a particularly important measure for career progression [1,11].

In dentistry, disparities between men and women researchers in both developed [12,13] and developing countries [14,15] have been show to exist, favouring men for funding and other financial support [7,16]. Men submit more research applications and receive more awards than women [17], hold more leadership positions in research projects [18], enjoy higher academic ranks [18,19], and are assigned more positions on dental journal editorial boards [20]. A citation gender gap is also observable [21]. Among the studies that have assessed the impact of programs and projects focused on gender inequality in various countries [5] and professions [19,22], none has yet attempted to evaluate the gender differences across Iranian dental faculties.

Iran has enjoyed a substantial increase in women's enrolment for higher education, with a significant increase in female graduates with higher education levels [23,24]. During the last decade, the number of female college graduates has increased substantially, leading to a higher percentage of female graduates than males [25]. However, men continue to occupy a higher number of editorial and administrative positions [26,27]. According to a previous national analysis on the research performance of Iranian medical academics in 2018, men had a higher median of citations per paper and papers per individual compared to women [28]. In Iran scientometric studies assessing gender ratios across dental schools within the country had not previously been undertaken. This study was carried out in 2021 in order to identify possible existing gender publication differences among dental faculty members based on scientometric indices. It includes speciality-specific differences as well across dental schools as identified by the data up to 8 August 2020.

## Methods

This was a cross-sectional scientometric study that reviewed the data of dental faculty members in Iran in 2020. We accessed Scopus outputs from the Iranian Scientometric Information Database (ISID, http://isid.research.ac.ir), Iran's Ministry of Health and Medical Education (MOHME) database that aims to aggregate the most up-to-date scientometric data of medical faculty members in Iran [28,29].

To have a more precise perspective on publication productivity, we excluded from the indices papers with a high number of authors ($\geq$100 authors, e.g. Global Burden of Disease studies) using a specific filter ("filter out high-author articles") on the ISID website.

### Variables

We assessed the following ISID variables: 1) Background characteristics: name, affiliation (dental school), university type, speciality, and academic rank; 2) Scientometric indicators: total number of papers, total number of citations, percentage of self-citation, h-index, g-index, citation per paper, number of papers in which the researcher was the first author, number of papers in which the researcher was the corresponding author, number of papers in which the researcher collaborated with international researchers, and year of the first published Scopus-indexed paper.

We calculated and added six other variables: gender (man or woman; manually done based on name; in cases of doubt, we contacted a student in that dental school to provide more information), university type (1 to 3; in which a lower figure indicates a higher-ranking institution, as determined by Iran's MOHME), number of years publishing (subtraction of the year of the first published Scopus-indexed paper from 2021), percentage of international papers (number of papers in which the researcher collaborated with international researchers divided by their total number of papers), percentage of first-author papers (number of papers in which the researcher was the first author divided by their total number of papers), percentage of corresponding-author papers (number of papers in which the researcher was the corresponding author divided by the total number of papers), and number of papers per year (number of papers divided by the number of years publishing).

### Data extraction and analyses

Data were extracted into a preformatted spreadsheet on 8 August 2020 by AS-M and then cleaned and verified by another author. Descriptive statistics were used to report the mean, median, standard deviation (SD), and range of selected variables. A men-to-women (MtoW) ratio was defined as the mean of each variable for men divided by the same mean for women.

Descriptive statistics for all the scientometric indices were reported using the median and inter-quartile range (IQR) in men and women for each specialty, dental school, academic rank, and university type. As our data were not normally distributed (confirmed by using the Shapiro–Wilk test), the Kruskal Wallis nonparametric test was used to examine the relationship between the background characteristics and scientometric indicators. The extracted data were analyzed using R v4.0.1 (2020-06-06) (R Core Team, R Foundation for Statistical Computing, Vienna, Austria. http://www.r-project.org). A No p-value level was considered statistically significant according to the statement by the American Statistical Association [30].

## Results

### Overall perspective

A total of 1850 dental faculty members appeared having publications in the ISID database, of which 1104 were women. Men had more papers (6583 vs. 6255) and citations (60410 vs. 39559). Among the 376 faculty members with zero papers, 234 (62.2%) were women. Almost half of women (N = 517) were in Type 2 universities, while nearly half of men (48.4%; N = 361) were affiliated with Type 1 universities. Fig 1 shows the MtoW ratio of the various dental faculties in Iran.

The median of most scientometric indices was higher in men than women (Table 1), except for the self-citation percentage, the international paper percentage, and the percentage of corresponding-author papers. In these three indices, women showed higher medians (P<0.001). Table 1 shows comparisons for a broader set of indices, providing greater detail for both genders.

| Type | School | MtoW | Type | School | MtoW | Type | School | MtoW | Type | School | MtoW |
|---|---|---|---|---|---|---|---|---|---|---|---|
| National | National | 0.68 | Type 2 | Urmia | 0.86 | | Golestan | 0.52 | | Bushehr | 1.00 |
| Type 1 | Tabriz | 1.30 | | Lorestan | 0.81 | | Shahrekord | 0.50 | | Bojnurd | 0.61 |
| | Mashhad | 1.14 | | Gilan | 0.78 | | Arak | 0.50 | | Kurdistan | 0.47 |
| | Shahid Beheshti | 1.05 | | Kermanshah | 0.72 | | Zanjan | 0.48 | | Ilam | 0.44 |
| | Isfahan | 0.79 | | Babol | 0.71 | | Hormozgan | 0.47 | | Qom | 0.41 |
| | Ahwaz | 0.77 | | Ardabil | 0.70 | | Zahedan | 0.35 | | Yasuj | 0.38 |
| | Tehran | 0.63 | | Yazd | 0.69 | | Birjand | 0.35 | | Alborz | 0.22 |
| | Mazandaran | 0.56 | | Hamadan | 0.68 | | Semnan | 0.26 | Misc | Shahed | 1.50 |
| | Shiraz | 0.50 | | Rafsanjan | 0.60 | | Qazvin | 0.26 | | | |
| | Kerman | 0.49 | | Kashan | 0.55 | Type 3 | Artesh | 3.25 | | | |

**Fig 1. Men-to-women (MtoW) ratio of dental faculties in different universities of medical sciences in Iran.** Type: University type; MtoW: Men-to-women ratio; Misc: Miscellaneous.

## Gender differences in academic ranks

Women in each academic rank had a higher total number of papers with fewer total citations. The data showed more publication activity, citations, and international collaboration among the higher ranking academic members for both genders (Table 2).

When comparing MtoW ratio between different academic ranks, from assistant professor to full professor, a decreasing trend was seen in the number of citations, h-index, citations per paper, and the number of papers per year. In contrast, the number of papers, working years, and corresponding-author papers percentage increased for women. The highest and lowest MtoW ratio belonged to the self-citation percentage of associate professors (2.00) and corresponding-author papers percentage of assistant professors (0). Full details are illustrated in S1 Table. Rates of first-authored papers and corresponding-author papers tended to favour women in all ranks over men. There were slight differences between genders in years publishing and the number of papers published per year. The complete set of results for these factors is presented in Table 2.

## Gender differences in university types

Men in Type 1 universities had the highest median in all indices except for the percentage of corresponding-author papers. The differences between genders in self-citation percentage, h-index, g-index, working years, and the number of papers per year were slight.

When comparing MtoW ratio between different university types, from Type 1 to Type 2, a decreasing trend was seen in citations per paper, whereas the number of papers per year increased. The highest and lowest MtoW ratio belonged to citations of Type 2 (2.00) and corresponding-author papers percentage of Type 2 (0). Full details are illustrated in S2 Table.

Women in Type 1 universities had the highest corresponding-author papers percentage (27.3%), followed by women in Type 2 universities (25%) and men in Type 1 universities (18.4%). The complete results of these analyses are found in Table 3.

## Gender differences in various dental schools

The national MtoW ratio among dental faculty members was 0.68 in Iran. Five dental schools in Iran with the highest MtoW ratios were the following: Artesh (3.25), Shahed (1.5), Tabriz (1.30), Mashhad (1.14), and Shahid Beheshti (1.05). On the other hand, Alborz (0.22), Qazvin (0.22), Semnan (0.26), Birjand (0.35), and Zahedan (0.35) dental schools had the lowest MtoW ratios.

**Table 1. Scietometric indices for each gender (median [inter-quartile range]).**

| Index | Both genders | Men | Women | P-value |
|---|---|---|---|---|
| **Papers** | 3 (7) | 3 (9) | 2 (6) | < 0.001 |
| **Citations** | 8 (40.75) | 13 (27.2) | 6 (63) | < 0.001 |
| **Self-citations (%)** | 0 (3) | 0 (3) | 0 (2) | 0.083 |
| **h-index** | 1 (3) | 2 (4) | 1 (3) | < 0.001 |
| **g-index** | 2 (5) | 2.5 (7) | 1 (4) | < 0.001 |
| **Citations per paper** | 4 (5.70) | 5 (5.93) | 3.33 (5.42) | < 0.001 |
| **Working years** | 8 (6) | 8 (7) | 7 (5) | < 0.001 |
| **International papers (%)** | 0 (11.8) | 0 (16.7) | 0 (7.5) | < 0.001 |
| **First author papers (%)** | 20 (50) | 24.5 (46.2) | 17.6 (50) | 0.122 |
| **Corresponding-author papers (%)** | 22.2 (50) | 15.4 (40) | 25 (50) | < 0.001 |
| **Number of papers per year** | 0.5 (0.83) | 0.5 (0.83) | 0.4 (0.86) | 0.006 |

**Table 2. Gender publication ratios across academic ranks (median [inter-quartile range]).**

| Index | Assistant professors | | | Associate professor | | | Full professors | | | P-values |
|---|---|---|---|---|---|---|---|---|---|---|
| | Both | Men | Women | Both | Men | Women | Both | Men | Women | |
| **Papers** | 2 (4) | 1 (4) | 2 (4) | 12 (10) | 11 (11) | 12 (9.5) | 23 (22) | 21 (22) | 25.5 (21.8) | 0.877, 0.873, 0.482 |
| **Citations** | 3 (16) | 4 (20) | 3 (14) | 61.5 (90.75) | 71 (100) | 54 (81.5) | 199 (193) | 200 (190) | 194 (215) | 0.037, 0.191, 0.897 |
| **Self-citations (%)** | 0 (0) | 0 (0) | 0 (0) | 1 (5) | 2 (5) | 1 (5.75) | 3 (5) | 3 (5) | 4 (5) | 0.588, 0.927, 0.501 |
| **h-index** | 1 (2) | 1 (2) | 1 (2) | 4 (3) | 4 (3) | 4 (3) | 8 (4) | 8 (4) | 8.5 (4.25) | 0.064, 0.383, 0.394 |
| **g-index** | 1 (3) | 1 (3) | 1 (3) | 7 (5) | 7 (6) | 6 (5) | 13 (6) | 13 (7) | 12.5 (6.25) | 0.098, 0.247, 0.864 |
| **Citations per paper** | 2.8 (5) | 3.33 (5.57) | 2.5 (4.31) | 5.33 (4.9) | 5.75 (5.1) | 4.88 (4.61) | 8.22 (4.66) | 8.55 (4.36) | 7.45 (4.97) | <0.001, 0.240, 0.248 |
| **Working years** | 6 (5) | 6 (4) | 6 (4) | 11 (4) | 11 (4.25) | 11 (4) | 14 (4) | 14 (3.25) | 13 (3) | 0.021, 0.887, 0.182 |
| **International papers (%)** | 0 (4.762) | 0 (12.5) | 0 (0) | 3.64 (12.5) | 5.88 (15.4) | 0 (10.8) | 7.69 (19.191) | 10 (22.3) | 6.67 (13.6) | 0.018, 0.104, 0.393 |
| **First author papers (%)** | 0 (36.842) | 0 (36.3) | 0 (37.3) | 37.5 (32.229) | 35.3 (30) | 42.5 (37.5) | 41.67 (26.708) | 40 (20.4) | 44.7 (29.2) | 0.699, 0.016, 0.058 |
| **Corresponding-author papers (%)** | 16.67 (50) | 0 (42.9) | 25 (50) | 25 (33.164) | 18.2 (30.8) | 30.4 (26.2) | 25.78 (25.714) | 24 (23.0) | 30.9 (31.1) | 0.001, <0.001, 0.151 |
| **Number of papers per year** | 0.33 (0.667) | 0.33 (0.667) | 0.33 (0.667) | 1.08 (0.925) | 1 (1.03) | 1.17 (0.753) | 1.71 (1.26) | 1.52 (1.27) | 1.82 (1.16) | 0.661, 0.768, 0.329 |

P-values order: Assistant professors, associate professors, full professors.

### Gender differences in specialties

The top five faculty members with the highest h-index, number of papers, and citations in endodontics and oral and maxillofacial surgery (OMFS) were all men. In contrast, the top faculty members with the highest h-index, number of papers, and citations in community oral health (COH), dental materials, oral medicine, orthodontics, oral and maxillofacial pathology (from now on: pathology), and restorative dentistry were women. The top women with the highest h-index had an overall rank of 15–20 (dental materials, pathology, orthodontics, and prosthodontics).

The MtoW ratio in the median h-index was lower than 1 in dental materials (0.42) and OMFS (0.5), whereas this ratio was equal to 2 in pediatric dentistry, periodontics, and orthodontics. The MtoW ratio in the median corresponding authorship percentage was lower than 1 in most fields, except for COH (1.13) and dental materials (1.13). The lowest ratios were seen in radiology (0), restorative dentistry (0), and pathology (0.3). The MtoW ratios in the median number of papers per year were highest in periodontics (1.93), paediatric dentistry (1.91), and restorative dentistry (1.79), whereas the ratios were lowest in dental materials (0.49), OMFS (0.70) and radiology (0.78). Other details regarding gender differences in specialities are depicted in Table 4 and S3–S7 Tables.

## Discussion

Overall, we found multiple cases of gender differences regarding publication performance among dental faculty members in Iran. The number of woman dental faculty members was approximately 48% higher than that of males. Additionally, women had a higher median

**Table 3. Gender differences based on university types (median [inter-quartile range]).**

| Index | Type 1 universities | | | Type 2 universities | | | Type 3 universities | | | P-values |
|---|---|---|---|---|---|---|---|---|---|---|
| | Both | Men | Women | Both | Men | Women | Both | Men | Women | |
| **Papers** | 7 (12) | 8 (13) | 6 (11) | 2 (4) | 1 (5) | 2 (4) | 1 (2) | 1 (2) | 1 (2) | 0.877, 0.873, 0.482 |
| **Citations** | 28 (89) | 39 (118) | 23 (70.8) | 3 (17) | 4 (25) | 2 (14) | 0 (9) | 0 (11) | 0 (9) | 0.037, 0.191, 0.897 |
| **Self-citations (%)** | 1 (4) | 1 (4) | 0 (4) | 0 (0) | 0 (1) | 0 (0) | 0 (0) | 0 (0) | 0 (0) | 0.588, 0.927, 0.501 |
| **h-index** | 3 (4) | 3 (5) | 2 (4) | 1 (2) | 1 (2) | 1 (2) | 0 (1) | 0 (1.5) | 0 (1) | 0.064, 0.383, 0.394 |
| **g-index** | 4 (8) | 5 (8) | 4 (7) | 1 (3) | 1 (3) | 1 (3) | 0 (2) | 0 (2) | 0 (2) | 0.098, 0.247, 0.864 |
| **Citations per paper** | 5.13 (5.83) | 6 (5.61) | 4.45 (5.63) | 2.69 (4.75) | 3.3 (5.75) | 2.4 (4.4) | 3 (5.9375) | 3.19 (5.84) | 2.88 (5.88) | <0.001, 0.240, 0.248 |
| **Years Publishing** | 10 (7) | 10 (7) | 9 (6) | 6 (5) | 6.5 (4.25) | 6 (5) | 5 (4) | 6 (4.75) | 5 (4) | 0.021, 0.887, 0.182 |
| **International papers (%)** | 0 (16.667) | 4.55 (20) | 0 (13.1) | 0 (0) | 0 (5.56) | 0 (0) | 0 (0) | 0 (26.1) | 0 (0) | 0.018, 0.104, 0.393 |
| **First author papers (%)** | 29.41 (43.809) | 30.3 (38.9) | 26.7 (50) | 0 (43.30357) | 0 (38.5) | 0 (50) | 0 (28.24675) | 0 (21.5) | 0 (28.6) | 0.699, 0.016, 0.058 |
| **Corresponding-author papers (%)** | 23.81 (42.857) | 18.4 (39.0) | 27.3 (47.5) | 20 (50) | 10.7 (50) | 25 (50) | 0 (50) | 0 (33.3) | 0 (50) | 0.001, <0.001, 0.151 |
| **Number of papers per year** | 0.77 (1.038) | 0.75 (1.04) | 0.778 (1) | 0.33 (0.727) | 0.333 (0.8) | 0.333 (0.667) | 0.2 (0.5) | 0.2 (0.5) | 0.2 (0.5) | 0.661, 0.768, 0.329 |

P-values order: Type 1 university, Type 2 university, Type 3 university.

percentage of corresponding-author papers (15.4 vs 25). These findings appear to show a positive step towards more balanced publication contributions by men and women dental faculty members in Iran. They may also indicate a continued disproportionate assignment of labour

**Table 4. Gender differences in each field (median [inter-quartile range]).**

| Field | h-index | | | Corresponding author % | | | Papers per year | | | MtoW |
|---|---|---|---|---|---|---|---|---|---|---|
| | Both | Men | Women | Both | Men | Women | Both | Men | Women | |
| **COH** | 3 (3.25) | 4 (2.5) | 3 (3) | 33.33 (57.14) | 35 (37.7) | 30.95 (50) | 0.55 (0.61) | 0.75 (0.77) | 0.53 (0.58) | 1.33, 1.13, 1.42 |
| **Dental Materials** | 5 (8.75) | 5 (0.5) | 12 (10.5) | 23.61 (21.5) | 25 (8.64) | 22.22 (31.46) | 1 (1.11) | 0.83 (0.21) | 1.69 (1.01) | 0.42, 1.13, 0.49 |
| **Endodontics** | 2 (4) | 3 (5.8) | 2 (4) | 21.83 (47.03) | 14.84 (33.33) | 29.81 (45.31) | 0.65 (0.95) | 0.76 (1.04) | 0.63 (0.83) | 1.5, 0.5, 1.21 |
| **OMFS** | 1 (3) | 1 (4) | 2 (2.5) | 10.24 (40) | 8.7 (41.18) | 12.5 (33.33) | 0.42 (0.82) | 0.4 (0.79) | 0.57 (0.76) | 0.5, 0.7, 0.7 |
| **Oral Medicine** | 2 (3) | 2 (2.5) | 2 (3) | 23.81 (50) | 23.81 (38.18) | 24.26 (50) | 0.67 (0.92) | 0.67 (0.68) | 0.68 (1.05) | 1, 0.98, 0.99 |
| **Orthodontics** | 1 (3) | 2 (3) | 1 (2) | 25 (50) | 23.3 (42.56) | 25 (52.78) | 0.57 (0.91) | 0.58 (1.13) | 0.54 (0.75) | 2, 0.93, 1.07 |
| **Pathology** | 2 (4) | 3 (4) | 2 (4) | 32.13 (50) | 10 (34.29) | 33.33 (56.25) | 0.67 (1.1) | 0.6 (1.48) | 0.72 (1.08) | 1.5, 0.3, 0.83 |
| **Pediatric Dentistry** | 1 (2) | 2 (3.25) | 1 (2) | 25 (50) | 16.23 (46.15) | 31.58 (50) | 0.33 (0.58) | 0.63 (0.78) | 0.33 (0.53) | 2, 0.51, 1.91 |
| **Periodontics** | 1 (3) | 2 (4) | 1 (3) | 23.08 (50) | 20 (50) | 25 (50) | 0.38 (1) | 0.58 (1.04) | 0.3 (0.75) | 2, 0.8, 1.93 |
| **Prosthodontics** | 1 (2) | 1 (2) | 1 (2) | 17.42 (50) | 12.5 (50) | 25 (50) | 0.39 (0.8) | 0.47 (0.72) | 0.35 (0.83) | 1, 0.5, 1.34 |
| **Radiology** | 1 (3) | 1.5 (3) | 1 (3) | 14.29 (50) | 0 (25.57) | 20 (50) | 0.5 (0.88) | 0.39 (0.8) | 0.5 (0.98) | 1.5, 0, 0.78 |
| **Restorative Dentistry** | 1 (2.25) | 1 (4) | 1 (2) | 16.03 (50) | 0 (20) | 25 (50) | 0.33 (0.75) | 0.5 (1.04) | 0.28 (0.67) | 1, 0, 1.79 |

MtoW: Men-to-Women ratio; COH: Community Oral Health; OMFS: Oral and Maxillofacial Surgery; Pathology: Oral and Maxillofacial Pathology; Radiology: Oral and Maxillofacial Radiology.

MtoW order: h-index, corrsponding author %, papers per year.

for women in conducting dental research. Hence, these findings alone do not indicate either a clear favouring or disfavoring of women compared to men in the dental sciences in Iran.

In contrast to some studies indicating unequal academic job opportunities for men and women [18,31,32], the opposite appears to have occurred in Iran for dentistry. In Iran, we have seen a strong openness to hiring women as dental faculty members, as shown by our data that now a significantly higher percentage of new hirings are women as compared to men. This may be an example of the reverse "Gender-Equality Paradox in Science, Technology, Engineering, and Mathematics Education" (STEM), suggesting that women in more democratic, equitable and developed countries being less likely to pursue STEM studies and careers. In a non-egalitarian country like Iran, women with a higher rate of gender discrimination may be willing to pursue the most straightforward way to financial independence. This trajectory usually leads through science, technology, engineering, and mathematics education and professions [34].

We found that men had a higher mean and median number of working years as researchers (nearly one year of difference). A likely factor contributing to this difference might be their type of working universities as well as developing trends in hiring. Approximately half of the men were affiliated with Type 1 universities, while women mainly were in Type 2 universities. Given that Type 1 universities usually require their tenured members to have a significant research background (including years of publishing), the higher number of women in Type 2 universities appears to indicate an employment pattern for dentistry faculty members in which women have generally occupied roles that are considered as "early-career." When researching the number of dental faculty members, the distribution of man to woman members noticeably differed among dental schools in Iran. While the national MtoW ratio was favourable (0.68) for women, more gender-mainstreaming policies are necessary to move towards gender equality in hiring academic staff in Iranian medical universities. Of course, this requires an existing set of infrastructure to build upon. For instance, Artesh University of Medical Sciences is affiliated with the military forces in Iran. It is thus not surprising to see in that university the highest MtoW ratio in its faculty members.

Women had more corresponding-author papers in all three academic ranks (assistant, associate, or full professor) than men in the same positions. In Iran, the published theses of master's and doctoral students in dental schools tend to assign supervisors as the corresponding authors. This may account for the high prevalence of woman corresponding-author papers: woman dental faculty members may tend to focus their research projects on supporting the education of their students. Additionally, the higher research labour that comes from these corresponding roles can limit their available time to engage in international collaborations because the thesis projects are generally conducted on a sub-national level and tend to focus on local issues. Our findings appear to substantiate this premise. Women were less likely to take part in international collaborations leading to publication, and men had a higher mean percentage of international papers. This could imply better international partnerships for men (12.30 vs. 9.57). These results align with previous studies indicating how gender disparities can occur in scholarly authorship [33].

In the "Global Gender Gap Report 2020" published by the World Economic Forum, among the 153 evaluated countries, Iran was ranked 148 (down from 142 in 2018) [34]. Although there have been enormous efforts in Iran over the past decades to advance women's participation in science [35], men still occupy the highest-ranking administrative positions in the top-ranked dental schools and in MOHME-supported institutions generally. According to the ISID platform, less than 20% of dental school deans in Iran were women (16.7%, 7 of 42) at the time of our study. We also found that less than one-third (7 of 22) of chief editorial positions in Iran's dental journals were occupied by women (data derived from the MOHME website,

journals.research.ac.ir). An earlier study noted similar gender differences among dental school faculty in the United States [36]. This can be due to different reasons such as work-life balance, stereotyping, lack of networking, family responsibilites, and limited access to resources.

Woman physicians in academia in Iran, on average, receive a lower salary compared with their male counterparts. A similar pattern is found in dentistry at all academic ranks [37,38]. Women faculty members also face are also less represented in awarded grant applications for research funding [39], unequal departmental resource allocation [40], and difficulty with obtaining the desired mentorship [41]. Women may also be disproportionately impacted by personal factors related to societal demands, such as greater responsibilities within the home [42]. A less positive work ambience for women in comparison with men in dental schools may also contribute to the differences in publication results, for example, receiving less respect from students, different expectations from patients in clinical procedures, and facing more difficulties in obtaining a work-life balance [43–45]. All these factors could reduce the publication productivity of woman faculty members. In our study, men were more productive, and published 328 more papers, contributing to nearly 55% of the publication output by dental faculties. However, the difference narrows to 26% when considering annual productivity. One large study found that while, on average male scientists publish 13.2 papers during their career, woman authors publish only an average of 9.6 throughout their career, resulting in a 27% gender gap in total productivity [32]. This is half the gap found in our study, which may suggest that women in Iran are not attracted to dental science production as much as in other parts of the world due to some structural barriers (e.g., their role in family support or potential discrimination they may face in the academic situation in Iran).

The OMFS and prosthodontics field specialties had the highest gender disparities regarding the number of women employed in those faculties. The number of woman dental faculty members was found to be highest in the pediatric dentistry specialty. The underrepresentation of women in OMFS [46,47] and their overrepresentation in pediatric dentistry [48] have been reported in other countries previously. One possible factor causing a notable gender gap in the OMFS field is that OMFS residency in Iran poses considerable stress and labour burden on residents. Moreover, the duration of specialty education conflicts with childbearing years and might be a potential reason for the specialist gender gap. The higher number of male dental faculty members in this field in the past may be self-perpetuating. On the other hand, pediatric dentistry appears to show less hidden or overt bias and greater acceptability for women [49].

Our research results show promise for an increasing amount of scholarly publication by women in dental faculties in Iran. The trend looks to be new and still faces opposition from a general secular trend in the country. Sustainable changes can only be expected when barriers to the full participation of women in the academic enterprise are reduced [38]. With a higher number of woman dental faculty members in Iran, it is expected to see an ongoing increase in the number of scholarly publications by woman dental faculty members. This is likely to be accompanied by a more general trend toward more opportunities for women to participate in leadership positions in administrative and editorial positions in the future of Iran's dental sciences. Increasing gender equality in dental publications in Iran contributes to an overall global goal of gender equality as a fundamental goal of education, science, and equity in society [50].

## Limitations

Our focus on scholarly publication data limits our analysis to appreciating gender differences within the single career measure of publication contributions. It does not take into account other career characteristics, including the career dynamics in which dental faculty in Iran participate: the important areas of teaching, administration, and contributions to the life of the

faculty and the school. In dentistry, contributions to research on medical devices and pharmaceuticals and patent developments are crucial for the development of the dental sciences within the country and internationally. A further limitation was that the Scopus database only includes English language publications, imposing a significant bias on results for a country where English is not the official language, and faculty members are accustomed to publishing in Persian and other international languages. Further research should provide broader, more needed evidence on the underlying factors influencing publication dispositions, career choices, international collaborations, hiring practices, and other gender trends in Iran's dental faculties.

## Conclusions

Whereas women occupy more positions in the dental faculties in Iran, men had an overall better scientometric performance. When categorizing based on university type and academic rank, however, the differences were reduced markedly. More research in this field should be encouraged and supported in order to better assess trends and underlying issues affecting gender differences among dental faculty members in Iran.

## Supporting information

**S1 Table. Number of dental faculty members in each specialty (\*: Ratio > 1).**
(DOCX)

**S2 Table. Dental faculty members' h-index by gender and specialty (SD) (\*: Lower than 1).**
(DOCX)

**S3 Table. Number of papers by gender and specialty (\*: Lower than 1).**
(DOCX)

**S4 Table. Citations per paper by gender and speciality (\*: Lower than 1).**
(DOCX)

**S5 Table. The percentage of first-author papers by gender and speciality (\*: Ratio < 1, ^: Ratio > 2).**
(DOCX)

**S6 Table. The percentage of corresponding-author papers by gender and speciality (\*: Ratio < 1, ^: Ratio > 2).**
(DOCX)

**S7 Table. Number of papers per year by gender and specialty.**
(DOCX)

## Acknowledgments

We wish to thank Dr Vasantha Muthuswamy (President, Forum for Ethics Review Committees in India [FERCI], Mumbai, India) for her valuable comments on our paper.

## Author Contributions

**Conceptualization:** Ahmad Sofi-Mahmudi.

**Data curation:** Ahmad Sofi-Mahmudi.

**Formal analysis:** Ahmad Sofi-Mahmudi, Erfan Shamsoddin, Lisa M. DeTora, Barbara E. Bierer, Francis P. Crawley.

**Investigation:** Ahmad Sofi-Mahmudi, Erfan Shamsoddin, Morenike Oluwatoyin Folayan, Ching Shan Lii, Francis P. Crawley.

**Methodology:** Ahmad Sofi-Mahmudi.

**Project administration:** Ahmad Sofi-Mahmudi.

**Resources:** Ahmad Sofi-Mahmudi.

**Software:** Ahmad Sofi-Mahmudi.

**Supervision:** Ahmad Sofi-Mahmudi.

**Validation:** Ahmad Sofi-Mahmudi.

**Visualization:** Ahmad Sofi-Mahmudi.

**Writing – original draft:** Ahmad Sofi-Mahmudi, Erfan Shamsoddin, Lisa M. DeTora, Barbara E. Bierer, Perihan Elif Ekmekci, Morenike Oluwatoyin Folayan, Ching Shan Lii, Marcos Roberto Tovani-Palone, Francis P. Crawley.

**Writing – review & editing:** Ahmad Sofi-Mahmudi, Erfan Shamsoddin, Lisa M. DeTora, Barbara E. Bierer, Perihan Elif Ekmekci, Morenike Oluwatoyin Folayan, Ching Shan Lii, Marcos Roberto Tovani-Palone, Francis P. Crawley.

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
