## [Decision Letter · Decision Letter 0]

29 Aug 2023

PONE-D-23-22007An Investigation into Gender Distributions in Scholarly Publications among Dental Faculty Members in IranPLOS ONE

Dear Dr. Sofi-Mahmudi,

Thank you for submitting your manuscript to PLOS ONE. After careful consideration, we feel that it has merit but does not fully meet PLOS ONE’s publication criteria as it currently stands. Therefore, we invite you to submit a revised version of the manuscript that addresses the points raised during the review process.

We look forward to receiving your revised manuscript.

Kind regards,

Yolanda Malele-Kolisa, BDS, MPH, MDent, PhD

Academic Editor

PLOS ONE

Journal Requirements:

Reviewers' comments:

Reviewer's Responses to Questions

**Comments to the Author**

1. Is the manuscript technically sound, and do the data support the conclusions?

Reviewer #1: Yes

2. Has the statistical analysis been performed appropriately and rigorously? 

Reviewer #1: Yes

3. Have the authors made all data underlying the findings in their manuscript fully available?

Reviewer #1: Yes

4. Is the manuscript presented in an intelligible fashion and written in standard English?

Reviewer #1: Yes

5. Review Comments to the Author

Reviewer #1: This paper brings a very important and recent theme. Authors investigated gender differences in several scientometric indices among dental researchers in Iran. The manuscript is well written, results are solid and an appropriate discussion was written. After the reviewing the paper, my comments are listed below:

Introduction

- Very informative and well written. No suggestion for this section.

Methods

- Do you think that "observation study" is the most appropriate definition for a scientometric study?

- Did you collect data about research funding and other financial support? How this could have impacted scientometrics?

Results

- It would be interesting to present names, positions and research area of the most prolific men and women Iranian authors.

- It would also be relevant if some of these researchers were interviewed to talk about their careers showing what, in their opinion, were the most important aspects that contributed for the position they have achieved.

Discussion

Adequate

6. PLOS authors have the option to publish the peer review history of their article (what does this mean?). If published, this will include your full peer review and any attached files.

Reviewer #1: No

---

## [Author Response · Author response to Decision Letter 0]

30 Aug 2023

Reviewer #1: This paper brings a very important and recent theme. Authors investigated gender differences in several scientometric indices among dental researchers in Iran. The manuscript is well written, results are solid and an appropriate discussion was written. After the reviewing the paper, my comments are listed below:

Introduction

- Very informative and well written. No suggestion for this section.

Authors: Thanks for your kind comments.

Methods

- Do you think that "observation study" is the most appropriate definition for a scientometric study?

Authors: Thanks for your comment. We agree with you. To prevent confusion with the typical definitions of “observational studies,” we changed the study design name to “cross-sectional scientometric.” Now it reads:

“This was a cross-sectional scientometric study that reviewed the data of dental faculty members in Iran in 2020.”

- Did you collect data about research funding and other financial support? How this could have impacted scientometrics?

Authors: Thanks for your comment. The ISID database does not have such kind of data, and there is a reason for that: Dental faculty members in Iran (unlike countries with a free market) very rarely get industrial funding. Based on our experience (ASM and ES) at the national granting body of Iran (NIMAD), they even rarely apply for funding from such institutions. The oral and dental research in Iran is mainly unfunded or funded locally by the university. Therefore, we do not think seeking funding sources will add much value to the study.

Results

- It would be interesting to present names, positions and research area of the most prolific men and women Iranian authors.

Authors: Thanks for your comment. We had already mentioned the top five men in the first sentence of the “Gender differences in specialties” subsection. However, we added the information about the top women, as follows:

“The top women with the highest h-index had an overall rank of 15–20 (dental materials, pathology, orthodontics, and prosthodontics).”

- It would also be relevant if some of these researchers were interviewed to talk about their careers showing what, in their opinion, were the most important aspects that contributed for the position they have achieved.

Authors: We appreciate your comment, and we think this is a very interesting idea. We also have mentioned the next steps in the last sentence of the “Limitations” subsection in the Discussion. We will consider this as our next project about gender inequalities in Iran.

Discussion

Adequate

Authors: Thanks for your kind comment.

---

## [Decision Letter · Decision Letter 1]

7 Nov 2023

PONE-D-23-22007R1An Investigation into Gender Distributions in Scholarly Publications among Dental Faculty Members in IranPLOS ONE

Dear Dr. Sofi-Mahmudi,

Thank you for submitting your manuscript to PLOS ONE. After careful consideration, we feel that it has merit but does not fully meet PLOS ONE’s publication criteria as it currently stands. Therefore, we invite you to submit a revised version of the manuscript that addresses the points raised during the review process.

We look forward to receiving your revised manuscript.

Kind regards,

Yolanda Malele-Kolisa, BDS, MPH, MDent, PhD

Academic Editor

PLOS ONE

Dear Authors

Thank you for the submission. Your manuscript went through the rigorous review process and independent reviewers. Both recommend improvement of the manuscript as suggested in the documents.

Kindky revise and resubmit.

Reviewers' comments:

Reviewer's Responses to Questions

**Comments to the Author**

1. If the authors have adequately addressed your comments raised in a previous round of review and you feel that this manuscript is now acceptable for publication, you may indicate that here to bypass the “Comments to the Author” section, enter your conflict of interest statement in the “Confidential to Editor” section, and submit your "Accept" recommendation.

Reviewer #2: (No Response)

2. Is the manuscript technically sound, and do the data support the conclusions?

Reviewer #2: Partly

3. Has the statistical analysis been performed appropriately and rigorously? 

Reviewer #2: I Don't Know

4. Have the authors made all data underlying the findings in their manuscript fully available?

Reviewer #2: Yes

5. Is the manuscript presented in an intelligible fashion and written in standard English?

Reviewer #2: No

6. Review Comments to the Author

Reviewer #2: 1. Abstract

- The background statement is an aim.

- A more suitable introductory paragraph for the background is required.

2. Methods

- Observationl study - should be more specific i.e., cross sectional.

- Describe what 'Scopus outputs' are.

- Variables

* Is the 'name' variable related to the name of the faculty members? If so, does the actual name of the author have any bearing on the analysis and study outcomes?

* 'Year of the first published Scopus indexed paper' - are papers which are not Scopus indexed not included in the study? If so, should this be stated as an exclusion criteria?

- Data sourcing and collection steps could be more descriptive - keywords to source dental papers and faculty members; justification for sourcing only Scopus-indexed papers. Were staff categorised into full/part time staff?

3. Results

- Overall perspective

* 1st paragraph is repeated in second paragraph, line 4.

* 2nd paragraph, line 4 : consider rephrasing to the opposite perspective e.g., over half of the women where in type 1 universities. Conversely, over half of the men were in type 2 universities.

- Table 2: Is index 'working years' the same as index 'years publishing'? index 'years' publishing' is used in Table 1&3.

- Section 'gender differences in specialities'

* paragraph 1, line 3: Do the top faculty members for COH, dent mat, oral med etc also have the highest 'h-index, number of papers and citations' as described in the previous sentence or is their high performance in other indices? The sentence needs to be more clear.

4. Discussion

- paragraph 1, line 9-11 may be better suited as a concluding statement.

- paragraph 2, line 4 "...indicated by the fact that now a significantly higher percentage of new hirings are women as compared to men." requires a reference

- paragraph 2, line 5-10 : gender equality paradox statement may require rephrasing. The concept speaks more to women in more democratic, equitable and developed countries being less likely to pursue STEM studies and careers. The deduction of the oppposite - women in non-egalitarian / inequatable countries are more likely to pursue careers in STEM - is not necessarily true.

- parapgrpah 3 line 4-5: the MtoW representation seems to be contradictory to the 1st paragraph in Results. The sentence states that almost 50% (i,e., less than half; 517/1104=47%) of women are in type 2 universities and 48.4% of men are in type 1 universities. This means there are more women (53% (100%-43%)) in type 1 uni's and more men in type 2 uni's. The footnote in the Methods section states that only results of type 1 and 2 universities were considered. Gender distrubution between type 1&2 universities requires clarification.

- paragraph 3 line 12: consider 'gender mainstreaming' instead of 'gender aware'.

- paragraph 4 line 5 - 9 statement needs a reference/s.

- paragrpah 5 - introduce paragraph as an illustration of the possible barriers and challenges that contribute to the gender distribution disparity in scholarly publications.

- paragraph 5, line 9-10 - referencing style inconsistent

- paragraph 6 line 13 - this may be incorrectly worded. Men did not publish nearly 55% more papers than women. Rather, men published 328 more papers than women, consequently contriubting to nearlyy 55% of the publication output by dental faculties.

- paragraph 6 line 13 - should these calculations be accounted for in the results section?

Typography and grammer

- Desription belown figure 1: MtoW: Male-to-female ratio - change to male-to-women ratio.

replace academic 'tier' with academic 'rank'

- Background paragraph 3, line 9 : assessing instead of accessing

- women instead of female (term consistency)

7. PLOS authors have the option to publish the peer review history of their article (what does this mean?). If published, this will include your full peer review and any attached files.

Reviewer #2: No

---

## [Author Response · Author response to Decision Letter 1]

1 Mar 2024

Reviewer #2: 1. Abstract

- The background statement is an aim.

- A more suitable introductory paragraph for the background is required.

Authors: Thanks for your comment. We added a background sentence to the abstract.

2. Methods

- Observationl study - should be more specific i.e., cross sectional.

Authors: Thanks. Amended.

- Describe what 'Scopus outputs' are.

Authors: We have already described the Scopus outputs in the variables section. Those included “total number of papers, total number of citations, percentage of self-citation, h-index, g-index, citation per paper, number of papers in which the researcher was the first author, number of papers in which the researcher was the corresponding author, number of papers in which the researcher collaborated with international researchers, and year of the first published Scopus-indexed paper.”

- Variables

* Is the 'name' variable related to the name of the faculty members? If so, does the actual name of the author have any bearing on the analysis and study outcomes?

Authors: The variable name includes the names of the faculty members. We used this variable to detect the gender of the faculties.

* 'Year of the first published Scopus indexed paper' - are papers which are not Scopus indexed not included in the study? If so, should this be stated as an exclusion criteria?

Authors: As we have mentioned in the first paragraph of the Methods, we only used the Scopus data. Therefore, it can be implied from the text that our focus is only on Scopus and not other sources such as Google Scholar.

- Data sourcing and collection steps could be more descriptive - keywords to source dental papers and faculty members; justification for sourcing only Scopus-indexed papers. Were staff categorised into full/part time staff?

Authors: Thanks for the comment. As indicated in the first paragraph of the Methods, the ISID is a formal database of the Ministry of Health of Iran. They have the records of all the faculty members, including those working in dental schools. Regarding using Scopus-indexed papers, it was the only option that we had and that database does not provide the records for other databases. Also, we do not have part-time dental faculties in Iran. They all are full-time.

3. Results

- Overall perspective

* 1st paragraph is repeated in second paragraph, line 4.

Authors: Thanks for the comment. We fixed that problem.

* 2nd paragraph, line 4 : consider rephrasing to the opposite perspective e.g., over half of the women where in type 1 universities. Conversely, over half of the men were in type 2 universities.

Authors: Thanks for your comment. We have three university types in Iran, and therefore, we cannot say that most of the women were affiliated with Type 1 universities. The highest percentage of affiliations for genders are the ones that are mentioned in the text (men: Type 1, women: Type 2).

- Table 2: Is index 'working years' the same as index 'years publishing'? index 'years' publishing' is used in Table 1&3.

Authors: Thanks for your keen comment. We changed “Years publishing” to “Working years.”

- Section 'gender differences in specialities'

* paragraph 1, line 3: Do the top faculty members for COH, dent mat, oral med etc also have the highest 'h-index, number of papers and citations' as described in the previous sentence or is their high performance in other indices? The sentence needs to be more clear.

Authors: Thanks for the comment. We added the criteria (just as described in the previous sentence).

4. Discussion

- paragraph 1, line 9-11 may be better suited as a concluding statement.

Authors: Thanks for your comment. We removed the sentence. The conclusion already had a similar sentence.

- paragraph 2, line 4 "...indicated by the fact that now a significantly higher percentage of new hirings are women as compared to men." requires a reference

Authors: Thanks for your comment. That sentence was based on the results of our study that the number of female faculty members is higher than men. We amended the sentence.

- paragraph 2, line 5-10 : gender equality paradox statement may require rephrasing. The concept speaks more to women in more democratic, equitable and developed countries being less likely to pursue STEM studies and careers. The deduction of the oppposite - women in non-egalitarian / inequatable countries are more likely to pursue careers in STEM - is not necessarily true.

Authors: Thanks for your keen comment. We rephrased the paragraph accordingly.

- parapgrpah 3 line 4-5: the MtoW representation seems to be contradictory to the 1st paragraph in Results. The sentence states that almost 50% (i,e., less than half; 517/1104=47%) of women are in type 2 universities and 48.4% of men are in type 1 universities. This means there are more women (53% (100%-43%)) in type 1 uni's and more men in type 2 uni's. The footnote in the Methods section states that only results of type 1 and 2 universities were considered. Gender distrubution between type 1&2 universities requires clarification.

Authors: Thanks for the comment. As mentioned in a previous comment (and also in the methods), we have three university types in Iran. In the footnote in the Methods we meant we included all the faculty members in the analyses but in the “Gender differences in university types” section, we only reported the results for type 1 and 2 universities. To avoid confusion, we removed the first sentence of the footnote.

- paragraph 3 line 12: consider 'gender mainstreaming' instead of 'gender aware'.

Authors: Thanks for your keen comment. Amended accordingly.

- paragraph 4 line 5 - 9 statement needs a reference/s.

Authors: Thanks for your comment. These statements are only our guesses for explaining “the high prevalence of female corresponding-author papers.” Therefore, we think we do not need any references for these sentences.

- paragrpah 5 - introduce paragraph as an illustration of the possible barriers and challenges that contribute to the gender distribution disparity in scholarly publications.

Authors: Thanks for your comment. We mentioned some of the barriers and challenges at the end of the paragraph.

- paragraph 5, line 9-10 - referencing style inconsistent

Authors: Thanks for your comment. Since it is not a reference rather a website to catch the data, we preferred to mention the web address in the text.

- paragraph 6 line 13 - this may be incorrectly worded. Men did not publish nearly 55% more papers than women. Rather, men published 328 more papers than women, consequently contriubting to nearlyy 55% of the publication output by dental faculties.

Authors: Thanks for your comment. We amended it accordingly.

- paragraph 6 line 13 - should these calculations be accounted for in the results section?

Authors: Thanks for the comments. Since it needs trivial calculations, we suppose that readers can do the calculations themselves.

Typography and grammer

- Desription belown figure 1: MtoW: Male-to-female ratio - change to male-to-women ratio.

Authors: Thanks for your comment. Amended.

replace academic 'tier' with academic 'rank'

Authors: Thanks for your comment. Amended.

- Background paragraph 3, line 9 : assessing instead of accessing

Authors: Thanks for your comment. Amended.

- women instead of female (term consistency)

Authors: Thanks for your comment. Amended.

---

## [Editor Report · Decision Letter 2]

5 Mar 2024

An Investigation into Gender Distributions in Scholarly Publications among Dental Faculty Members in Iran

PONE-D-23-22007R2

Dear Dr. Sofi-Mahmudi,

We’re pleased to inform you that your manuscript has been judged scientifically suitable for publication and will be formally accepted for publication once it meets all outstanding technical requirements.

Kind regards,

Yolanda Malele-Kolisa, BDS, MPH, MDent, PhD

Academic Editor

PLOS ONE
---

## [Editor Report · Acceptance letter]

23 Mar 2024

PONE-D-23-22007R2 

PLOS ONE

Dear Dr. Sofi-Mahmudi, 

I'm pleased to inform you that your manuscript has been deemed suitable for publication in PLOS ONE. Congratulations! Your manuscript is now being handed over to our production team.

Kind regards, 

on behalf of

Prof Yolanda Malele-Kolisa 

Academic Editor

PLOS ONE